# Synthesis and PET Imaging Biodistribution Studies of Radiolabeled Iododiflunisal, a Transthyretin Tetramer Stabilizer, Candidate Drug for Alzheimer’s Disease

**DOI:** 10.3390/molecules29020488

**Published:** 2024-01-18

**Authors:** Sameer M. Joshi, Thomas C. Wilson, Zibo Li, Sean Preshlock, Vanessa Gómez-Vallejo, Véronique Gouverneur, Jordi Llop, Gemma Arsequell

**Affiliations:** 1CIC BiomaGUNE, Basque Research and Technology Alliance (BRTA), Paseo Miramón 182, Parque Tecnológico de San Sebastián, 20009 Donostia-San Sebastián, Spain; sameerjoshi94@gmail.com (S.M.J.); vgomez@cicbiomagune.es (V.G.-V.); 2Department of Radiology, The University of North Carolina at Chapel Hill, Chapel Hill, NC 27599, USA; zibo_li@med.unc.edu; 3Chemistry Research Laboratory, Oxford University, Oxford OX1 3TA, UK; tcwilson1610@gmail.com (T.C.W.); veronique.gouverneur@chem.ox.ac.uk (V.G.); 4Lineberger Comprehensive Cancer Center, The University of North Carolina at Chapel Hill, Chapel Hill, NC 27514, USA; 5Institut de Química Avançada de Catalunya (IQAC), Spanish National Research Council (CSIC), 08034 Barcelona, Spain

**Keywords:** iododiflunisal, transthyretin tetramer stabilizer, small-molecule chaperone, amyloid beta, in vivo imaging, ^18^F, positron emission tomography (PET)

## Abstract

The small-molecule iododiflunisal (IDIF) is a transthyretin (TTR) tetramer stabilizer and acts as a chaperone of the TTR-Amyloid beta interaction. Oral administration of IDIF improves Alzheimer’s Disease (AD)-like pathology in mice, although the mechanism of action and pharmacokinetics remain unknown. Radiolabeling IDIF with positron or gamma emitters may aid in the in vivo evaluation of IDIF using non-invasive nuclear imaging techniques. In this work, we report an isotopic exchange reaction to obtain IDIF radiolabeled with ^18^F. [^19^F/^18^F]exchange reaction over IDIF in dimethyl sulfoxide at 160 °C resulted in the formation of [^18^F]IDIF in 7 ± 3% radiochemical yield in a 20 min reaction time, with a final radiochemical purity of >99%. Biodistribution studies after intravenous administration of [^18^F]IDIF in wild-type mice using positron emission tomography (PET) imaging showed capacity to cross the blood-brain barrier (ca. 1% of injected dose per gram of tissue in the brain at t > 10 min post administration), rapid accumulation in the liver, long circulation time, and progressive elimination via urine. Our results open opportunities for future studies in larger animal species or human subjects.

## 1. Introduction

Alzheimer’s disease (AD), the most common cause of dementia affecting over 13 million people worldwide, is characterized by the accumulation of amyloid-β (Aβ) aggregates, [1,2] the appearance of hyperphosphorylated Tau protein (p-Tau), [3] synaptic dysfunction, [4] neuroinflammation, [5] structural cerebrovascular alterations, and early deficits in cerebral glucose uptake and cerebral blood flow responses [6]. 

Transthyretin (TTR), a naturally occurring endogenous homotetrameric protein that is mainly synthesized by the liver and the choroid plexus (CP) and secreted to blood and cerebrospinal fluid (CSF), respectively, acts as a transporter protein of the thyroid hormone thyroxine (T_4_), as well as retinol. TTR represents 20% of the total CSF protein [7]. TTR levels are decreased in cerebrospinal fluid [8,9] and plasma [10,11,12] of AD patients. AD mice with genetic TTR reduction (AD/TTR+/−) show increased Aβ levels and deposition compared to AD/TTR+/+ [13], whereas overexpressing human WT TTR in an AD mouse model decreases neuropathology and Aβ deposition [14]. TTR has been proven to play an important role in AD pathogenesis, both preclinically and clinically [15]. Still, the mechanism underlying the protective role of TTR in AD remains unresolved, although some studies have shed light on this process. Several in vitro studies suggest that TTR binds Aβ, avoiding its aggregation and toxicity [16,17,18]. Additionally, TTR promotes Aβ brain efflux through the blood-brain barrier (BBB) [19] and increases the expression of low-density lipoprotein receptor-related protein 1 (LRP1) the main brain Aβ efflux receptor [20]. Recently, it has been demonstrated that TTR also regulates vascular function [21] and is involved in neuronal physiology by increasing neurite outgrowth [22] and impacting on axonal transport, [23] suggesting that TTR may enhance neuroprotection.

A noteworthy detail is that TTR is unstable. TTR stabilization can be achieved by means of small-molecule tetrameric stabilizers [24,25] such as: tolcapone, an approved drug for the treatment of Parkinson’s disease and repurposed for TTR amyloidogenesis [26]; tafamidis, an approved drug for the treatment of TTR amyloid cardiomyopathy [27]; the non-steroidal anti-inflammatory drug (NSAID) diflunisal (Figure 1, DIF) [28]; and other small molecules such as the investigational drug iododiflunisal (Figure 1, IDIF), an iodinated analogue of the NSAID diflunisal [29] that has proven to enhance TTR activity and improve AD-like pathology in AD mice [30,31]. These results have positioned IDIF as a potential disease-modifying drug for AD [32]. 

From a radiochemical point of view, the investigational drug Iododiflunisal (IDIF) is an interesting molecule because it contains both fluorine and iodine atoms in its chemical structure. Hence, labeling with radioiodine or radiofluorine, which are both commonly used in the clinical setting, can be achieved without modifying the chemical structure. In previous work, it has been shown that the formation of the TTR-IDIF complex increases blood-brain-barrier (BBB) permeability in mice [33]. This was proven by ^131^I-radioiodination of NSAID diflunisal (DIF) via electrophilic aromatic iodination using Na[^131^I]I in the presence of the oxidant chloramine-T, following a well-established method widely described in the literature [34,35]. This synthetic approach resulted in decay-corrected radiochemical yields close to 20%, and a radiochemical purity of the labeled compound exceeding 95% at the end of the synthesis. This yield was sufficient to undertake ex vivo/in vivo experiments in rodents using dissection and gamma counting, as well as single photon emission computerized tomography (SPECT) imaging, respectively. However, SPECT has limitations in terms of temporal resolution, sensitivity, and absolute quantification. In this regard, the development of radiolabeling strategies using positron emitters is highly desirable due to the unparalleled sensitivity and quantification opportunities of positron emission tomography (PET) imaging.

Here, we report the radiolabeling of IDIF with ^18^F-fluoride, as well as in vivo biodistribution studies of ^18^F-labeled IDIF ([^18^F]IDIF) in wild-type (WT) mice.

## 2. Results and Discussion

### 2.1. Synthesis of [^18^F]IDIF

The incorporation of ^131^I into IDIF was enabled in vivo studies using SPECT [33]. However, this radionuclide has limitations including low availability, difficult on-site production, and its approximately 90% beta emission, which may lead to radiotoxic effects. Additionally, it is well known that SPECT has limitations when compared to PET in terms of temporal resolution, sensitivity, and quantification. Radiolabeling with a widely available positron emitter would be desirable to enable in vivo imaging using PET. Among all positron emitters, ^18^F has favorable emission properties (almost 100% positron emission; an appropriate half-life of 109.7 min; a short positron range; and straightforward production with biomedical cyclotrons). Therefore, it was selected for our next labeling studies.

We considered isotopic exchange (^18^F/^19^F) to introduce ^18^F onto IDIF [36] because this method was expected to be straightforward and may be performed on the desired molecule without the need for preparing precursors. However, this approach has four main drawbacks. First, the presence of an iodine atom in the aromatic ring may lead to the ^18^F/I exchange, but literature reports indicate that the efficiency of aromatic F-F exchange is significantly higher than that of F-I exchange [37]; second, the absence of activated residues on the phenyl ring containing the fluorine atoms suggests the likely need for harsh reaction conditions and low [^18^F] incorporation ratios; third, it is well known that isotopic exchange reactions result in low molar activity; finally, the isotopic exchange reaction may occur in two different positions, leading to two different regioisomers. Still, the last two were not considered to be critical in the context of this work.

Anticipating potential problems in the isotopic (^18^F/^19^F) exchange reaction due to the presence of the alcohol and carboxylic acid groups on the phenyl ring, we first studied the isotopic exchange reaction on a protected IDIF derivative **4** (Figure 2A).

The protected IDIF derivative **4** was prepared in two steps from commercially available DIF (**1**) (Figure 3). First, DIF (**1**) was treated with potassium carbonate and methyl iodide in dimethylformamide (DMF) at room temperature (RT) to yield the desired *O*-alkylated product **3** with close-to-quantitative isolated yield (94%) (see Appendix A for NMR characterization). Reaction of compound **3** with iodine and [bis(trifluoroacetoxy)iodobenzene] in dichloromethane (DCM) at RT yielded compound **4** in excellent yield (85%) (see Appendix A for NMR characterization). Finally, the demethylation of compound **4** was accomplished using boron tribromide affording IDIF in 83% yield (See Appendix A for NMR characterization). This step was assayed because it could be applied as a final step in the radiosynthesis of [^18^F]IDIF.

Firstly, the ^18^F-radiofluorination was investigated with the crude reaction mixture being analyzed by high performance liquid chromatography (HPLC) coupled to a radioactive detector (radio-HPLC) to monitor the formation of the desired labeled compound and determine radiochemical conversion (RCC) values. When compound **4** (5 mM and 55 mM) was reacted with [^18^F]TBAF (produced as reported previously [38]) in DMSO or *N*-methyl-2-pyrrolidone (NMP) as the solvent at 140 °C (Figure 2A), the desired ^18^F-labeled product was not observed (see Appendix A). When the temperature was increased to 180 °C for 30 min in NMP (concentration of **4** = 55 mM), multiple radiofluorinated products were formed along with [^18^F]**4**, as confirmed by radio-HPLC (see Appendix A). Still, the low radiochemical conversion, as well as the need for purification and subsequent hydrolysis to reach [^18^F]IDIF, may lead to very low radiochemical yields. Hence, this route was abandoned.

Next, the reaction of IDIF with [^18^F]KF was carried out in the presence of potassium carbonate and Kryptofix^®^ 2.2.2 as the phase transfer catalyst. In all cases, reaction time was fixed at 20 min. As can be seen in Table 1, no formation of the product could be observed when 1 mg of IDIF was dissolved in 200 µL of DMSO (final concentration = 13 mM) and the reaction was conducted at 80 °C. The same results were obtained when 4 mg of IDIF were used in the same volume (53 mM). When the reaction temperature was increased to 120 °C, the formation of the desired labeled product could be detected, although RCC values were extremely low. Raising the reaction temperature to 160 °C resulted in RCC values of 12.5 ± 2.9 and 20.6 ± 3.7 for 1 and 4 mg of IDIF, respectively (see Supporting information, Appendix A for representative chromatograms). Interestingly, no radiolabeled impurities were observed in the HPLC profiles. The reaction conducted in DMF did not improve these results; hence, the conditions 1 mg of IDIF (to increase molar activity), DMSO as the solvent, and 160 °C were selected as the most appropriate to tackle in vivo experiments.

The whole automated synthesis, including purification by radio-HPLC (see Appendix A for representative chromatogram) and solid-phase extraction cartridge-based purification, allowed for the preparation of the tracer in >99% radiochemical purity (see Appendix A), 7 ± 3% decay corrected radiochemical yield, and with molar activity values of 0.15–0.35 GBq/µmol at the end of the synthesis (ca. 80 min overall production time).

### 2.2. In Vivo Imaging

Radiochemical yields enabled the execution of in vivo imaging studies using PET. The biodistribution of [^18^F]IDIF was investigated in wild-type mice. Animals (N = 3) were intravenously administered with the labeled compound, and dynamic whole-body PET images were acquired immediately after administration. Representative images registered at different times after administration (Figure 4a) confirm a rapid accumulation of the radioactivity in the liver, long residence of the radioactivity in blood, and progressive elimination via urine.

Quantification of the images (Figure 4b) confirmed that the labeled IDIF rapidly accumulated in the liver, with values of 27.5 ± 7.2% ID/g at ca. 7 min after administration, with a progressive decrease afterwards to reach a plateau at ca. 25% ID/g. The concentration of radioactivity in the volume of interest (VOI) drawn at the heart, which can be considered a surrogate of the concentration of radioactivity in the blood, confirms long circulation time. The presence of radioactivity in the kidneys and a progressive increase in the bladder confirm partial elimination via urine. The presence of radioactivity could also be detected in the lungs (12.6 ± 2.1% ID/g immediately after the start of image acquisition to reach a plateau at ca. 5% ID/g) and the brain (ca. 1%ID/g a t > 10 min after administration). However, these are highly irrigated organs, and these values could be overestimated because of the high concentration of radioactivity in blood.

The values obtained in our biodistribution studies show a good correlation with those previously obtained in dissection and gamma counting studies [33] (Figure 4c). Values obtained for the kidneys, liver, and heart show a good in vivo/ex vivo correlation. Because the previous studies were performed with [^131^I]IDIF, the result suggests a good stability of the labeled compounds after administration. The discrepancies observed in the bladder can be explained because in the studies reported in this work, the whole biodistribution study is performed under anesthesia, hence the animal does not urinate over the whole image acquisition period. However, in dissection and gamma counting, animals are maintained awake between injection and sacrifice. Differences observed in the lungs (higher values are obtained in vivo) can be explained by spill-over from the surrounding organs (liver and heart).

## 3. Materials and Methods


**General:**


All chemicals obtained commercially were of analytical grade and used without further purification. The NSAID Diflunisal was purchased from Combi-Blocks. [Bis(trifluoroacetoxy)iodo]benzene [(CF_3_CO_2_)_2_IC_6_H_5_] was purchased from Chem-Impex International; Iodomethane [CH_3_I], boron tribromide solution (BBr_3_), iodine, sodium sulfite, and trifluoroacetic acid (HPLC grade) were purchased from Sigma-Aldrich. Potassium carbonate, hydrochloric acid, sodium chloride, dimethylformamide (anhydrous), dimethyl sulfoxide (anhydrous), hexane, ethyl acetate, dichloromethane and methanol, acetonitrile (HPLC grade), and water (HPLC grade) were purchased from Fisher Scientific and used without further purification.

All reactions under non-radioactive conditions were carried out in oven-dried glassware under a positive pressure of argon or nitrogen unless otherwise mentioned with magnetic stirring. Air sensitive reagents and solutions were transferred via syringe or cannula. Reactions were monitored by analytical thin layer chromatography (TLC) on 0.25 mm pre-coated Merck 60 F254 silica gel plates (Merck KGaA, Darmstadt, Germany). Visualization was accomplished with either UV light (254 nm) or iodine adsorbed on silica gel. The ^1^H, ^19^F, and ^13^C NMR spectra of small molecules were recorded using on Agilent 400 MHz NMR Spectrometer. NMR spectra were recorded using CDCl_3_ as solvent and TMS as internal reference [^1^H NMR: CDCl3 (7.27); ^13^C NMR: CDCl_3_ (77.00)]. The peak assignment of product was performed using ChemDraw Professional V.16 software package. Data are reported as follows: chemical shift, multiplicity (s = singlet, d = doublet, t = triplet, m = multiplet, and br = broad), integration, and coupling constants (Hz). Column chromatographic separations were carried out on silica gel (60–120 mesh and 230–400 mesh).


**Chemistry**


**Synthesis of methyl 2’,4’-difluoro-4-methoxy-[1,1’-biphenyl]-3-carboxylate (3).** To a stirred suspension of commercially available diflunisal (**1**, 500 mg, 2 mmol, 1 eq) in a 50 mL round bottom flask, potassium carbonate (1.10 g, 8 mmol, 4 eq) was added at room temperature (RT) and placed under nitrogen. The reaction vessel was then evacuated and backfilled with nitrogen gas three times. Dry dimethyl formamide (DMF; 5 mL) was then added at RT. Methyl iodide (568 mg/249 μL, 4 mmol, 2 eq) was added after 10 min, and the reaction was stirred vigorously for 15 h. Water (20 mL) was then added, followed by extraction with ethyl acetate (EtOAc; 3 × 20 mL). The combined organic extracts were washed with 1% aq. HCl (30 mL), brine (30 mL), and sodium thiosulfate (30 mL), dried over MgSO_4_, filtered, and concentrated under vacuum. Silica gel column chromatography (15% EtOAc in hexane) yielded the desired compound **3** as a white solid (523 mg, isolated yield: 94%). ^1^H NMR (400 MHz, CDCl_3_, 23 °C, δ): δ 7.94 (s, 1H), 7.61 (d, *J* = 5.0 Hz, 1H), 7.38 (q, *J* = 5.0 Hz, 1H), 7.04 (d, *J* = 10 Hz, 1H), 6.97–6.88 (m, 2H), 3.95 (s, 3H), 3.91 (s, 3H); ^13^C NMR (100 MHz, CDCl_3_, 23 °C, δ): δ 172.5, 166.4, 158.7, 133.9, 132.0, 131.0, 127.0, 120.2, 112.2, 111.7, 111.5, 104.7, 104.1, 56.2, 52.1; ^19^F NMR (400 MHz, CDCl_3_, 23 °C, δ): –δ −111.5, −113.8; HRMS *m*/*z* calculated for C_15_H_13_O_3_F_2_ (M+H+): 279.0755; Found: 279.08253.

**Synthesis of methyl 2’,4’-difluoro-5-iodo-4-methoxy-[1,1’-biphenyl]-3-carboxylate (4).** To a stirred solution of methyl 2’,4’-difluoro-4-methoxy-[1,1’-biphenyl]-3-carboxylate (**3**) (300 mg, 1 mmol, 1 equiv.) in anhydrous dichloromethane (DCM; 12 mL) at 0 °C, [bis(trifluoroacetoxy)iodo] benzene (556 mg, 1.2 mmol, 1.2 eq) and iodine (274 mg, 1 mmol, 1 eq) were added. After 15 min, the reaction was allowed to warm to room temperature and allowed to stir for 3 h under darkness. Sodium sulfite (1 M aq. solution, 20 mL) was added, and the layers were separated. The organic phase was washed successively with water (20 mL) and brine (20 mL), dried over MgSO_4_, filtered, and concentrated under vacuum. Silica gel column chromatography (3% EtOAc in hexane) yielded compound 4 as a yellow oil (369 mg, isolated yield: 85%). TLC (20% EtOAc in hexane) ^1^H NMR (400 MHz, CDCl_3_, 23 °C, δ): δ 8.08 (s, 1H), 7.92 (s, 1H), 7.37 (q, *J* = 5.0 Hz, 1H), 6.98–6.89 (m, 2H), 3.96 (s, 3H), 3.94 (s, 3H); ^13^C NMR (100 MHz, CDCl_3_, 23 °C, δ): δ 165.3, 163.9, 158.8, 143.4, 132.5, 132.3, 131.2, 125.4, 111.9, 111.7, 104.6, 104.3, 94.1, 62.4, 52.6; ^19^F NMR (400 MHz, CDCl_3_, 23 °C, δ): –δ −109.8, −113.2; HRMS *m*/*z* calculated for C_15_H_12_O_3_F_2_I (M+H+): 404.9721; Found: 404.97919.

**Synthesis of iododiflunisal (2):** For a solution of compound **4** (70 mg, 0.17 mmol, 1.0 eq.) in DCM (2 mL), BBr_3_ (0.16 mL in 1 M solution in CH_2_Cl_2_) was added at −78 °C under nitrogen atmosphere. The reaction mixture was stirred at −78 °C for 15 min. Then, the reaction mixture was stirred at room temperature under nitrogen for 18 h. The reaction was quenched with a saturated sodium bicarbonate solution (10 mL; final pH = 9–10) and DCM was evaporated. Non-polar impurities were extracted with hexane (3 × 20 mL). The desired compound was extracted with DCM (3 × 20 mL), and the combined organic layers were washed with brine (1 × 10 mL), dried over MgSO_4_, and concentrated under vacuum to yield compound **2** as a white solid (54 mg, isolated yield: 83%). TLC (10% MeOH in dichloromethane). ^1^H NMR (400 MHz, DMSO-d_6_, 23 °C, δ): δ 7.82 (t, 1H), 7.76 (t, 1H), 7.49 (q, 1H), 7.26 (t, 1H), 7.11 (t, 1H); ^13^C NMR (100 MHz, DMSO-d_6_, 23 °C, δ): δ 172.5, 170.8, 164.1, 140.9, 131.7, 130.8, 124.5, 123.4, 119.6, 112.5, 112.2, 104.5, 87.6; ^19^F NMR (400 MHz, DMSO-d_6_, 23 °C, δ): –δ −112.4, −114.1. HRMS: (*m*/*z*) calculated for C_13_H_8_O_3_F_2_I (M+H^+^): 376.9408; Found: 376.94793, in accordance with our published data [29,39]. Furthermore, IDIF (**2)** was prepared in three synthetic steps with 66% overall yield.


**Radiochemistry**


All procedures involving the handling of radioactive substances were carried out in a radiochemistry laboratory with the required conditions of radiological protection and safety.


**Synthesis of [^18^F]4**


Anhydrous [^18^F]TBAF was obtained as previously reported [38]. To a stirred suspension of methyl 2’,4’-difluoro-5-iodo-4-methoxy-[1,1’-biphenyl]-3-carboxylate **2** (1.12 mg) placed in a V-vial containing a magnetic stirrer bar, dry 1-Methyl-2-pyrrolidinone (NMP-50 µL, 55 mM) and Anhydrous [^18^F]TBAF (ca. 4 MBq, 10 µL MeCN approx.) were added, and reaction mixture was heated at 180 °C for 30 min. Aliquot were extracted for HPLC analysis using a Phenomenex Kinetex^®^ (5 μm F5 100 Å, 250 × 4.6 mm) column as the stationary phase, and 0.1%TFA 95%water in acetonitrile (A)/0.1%TFA 95%acetonitrile in water (B) as the mobile phase, with the following gradient: 0 to 2 min: 50% B; 2 to 22 min: from 50% to 95% B; 22 to 27.1 min: 95% B; 27.1 to 28 min: from 95% to 5% B; 30 to 35 min: 5% B. Flow rate: 1 mL/min.

**Synthesis of [^18^F]IDIF:** Radiofluorination reactions were carried out using a fully automated synthesis module (TRACERlab FXFN, GE Healthcare). Fluorine-18 (^18^F) was generated in an IBA Cyclone 18/9 cyclotron by irradiation (target current = 44 µA) of ^18^O-enriched water with high energy (18 MeV) protons via ^18^O(p, n)^18^F reaction. For optimization runs, very short irradiations were performed (integrated current of 0.1 µAh in the target). For syntheses devoted to in vivo studies, irradiation was performed to achieve a final amount of radioactivity of 3.7 GBq, according to theoretical production yields. In all cases, [^18^F]F^-^ was trapped on a pre-conditioned Sep-Pak^®^ Accell Plus QMA Light cartridge (Waters, Milford, MA, USA) and then eluted to the reactor with a solution of 3.5 mg of Kryptofix K_2.2.2_/15 mg K_2_CO_3_ in a mixture of 500 mL of water and 1 mL of acetonitrile. After azeotropic evaporation of the solvent, a solution of IDIF (**2**) in the appropriate solvent was added and the reaction was carried out. During optimization of the experimental conditions, the reactor crude was flushed into a vial and the determination of radiochemical conversion was carried by radio-HPLC, using an Agilent 1200 Series system equipped with a UV-Vis variable wavelength detector (λ = 254 nm) and a radioactivity detector (Gabi, Raytest) connected in series. A Mediterranean C18 column (4.6 × 250 mm, 5 µm; Teknokroma, Spain) was used as the stationary phase, and aqueous NaH_2_PO_4_ solution (0.05 M pH 3.4; A)/acetonitrile (B) was used as the mobile phase at a flow rate of 1 mL/min, with the following gradient: t = 0–4 min, 90% A; t = 4–12 min, from 90% A to 40% A; t = 12–20 min, 40%A; t = 20–21 min, from 40% A to 90% A; t = 21–25 min, 90% A. For synthesis devoted to animal experiments, the reaction was carried out in DMSO (160 °C, 20 min). After cooling to 40 °C, the crude was purified by semi-preparative HPLC using a Nucleosil^®^ 100-7 column (10 × 250 mm, 5 µm; Macherey-Nagel^TM^) as the stationary phase and aqueous NaH_2_PO_4_ solution (0.05 M, pH 3.4)/acetonitrile (40/60) as the mobile phase at a flow rate of 4 mL/min (retention time = 11.5 min). The desired product was diluted with purified water (20 mL) and the radiotracer was retained on a C-18 cartridge (Sep-Pak^®^ Light, Waters, Milford, MA, USA) and further eluted with ethanol (0.5 mL). The ethanol solution was finally reconstituted with saline solution (4.5 mL). Chemical and radiochemical purity were determined by radio-HPLC using the same analytical method as above.


**Small animal PET imaging**


**General aspects.** Male C57BL/6 mice (10–11 weeks old, Charles River Laboratories) were used. The animals were maintained and handled in accordance with the Guidelines for Accommodation and Care of Animals (European Convention for the Protection of Vertebrate Animals Used for Experimental and Other Scientific Purposes) and internal guidelines. All experimental procedures were approved by ethical committee of CIC biomaGUNE and local authorities (authorization number: PRO-AE-SS-207) before conducting experimental work. Animals were housed in ventilated cages and fed on a standard diet ad libitum.

**PET image acquisition.** PET scans were performed using an eXplore Vista PET-CT camera (GE Healthcare, Waukesha, WI, USA). Scans were performed in mice anaesthetized with 4% isoflurane and maintained by 2–2.5% of isoflurane in 100% O_2_. Animals were placed into a mouse holder compatible with the PET acquisition system, and maintained normothermia using a water-based heating blanket at 37 °C. To ensure animal welfare, temperature and respiration rate were continuously monitored while they remained in the PET camera using a SAII M1030 system (SA Instruments, New York, NY, USA). The radiotracer (ca. 1.8 MBq; 100 µL) was injected in one of the lateral tail veins and PET scans were started immediately after (time gap of ca. 30 s). Whole body dynamic images were acquired in two bed positions for 29 frames (4 × 10, 4 × 20, 4 × 30, 4 × 60, 3 × 120, 3 × 240, 3 × 480, 4 × 600 s) for 90 min in the 400–700 keV energetic window. After each PET scan, CT acquisitions were also performed (140 mA intensity, 40 kV voltage) in order to provide anatomical information of each animal as well as the attenuation map for the later PET image reconstruction. Dynamic and static acquisitions were reconstructed (decay and CT-based attenuation corrected) with filtered back projection (FBP) using a Ramp filter with a cutoff frequency of 0.5 mm-1.

**Image analysis.** PET images were analyzed using PMOD image analysis software (Version 3.5, PMOD Technologies Ltd., Zurich, Switzerland). Volumes of interest (VOIs) were manually drawn on the CT images and translated to the PET images, and the concentration of radioactivity in each VOI was determined as a function of time. Values were finally expressed as percentage of injected dose per cubic centimeter (% ID/cm^3^).

## 4. Conclusions

In this work, we demonstrate that the small molecule IDIF, a TTR tetramer stabilizer and potential disease modifying drug for AD, can be efficiently radiolabeled with the positron emitter ^18^F via straightforward isotopic exchange (^18^F/^19^F) reaction, with reasonable radiochemical yields. Biodistribution studies in mice after intravenous administration confirmed capacity to cross the blood-brain barrier and long circulation time. These results open opportunities for future studies in larger animal species or human subjects and may aid in the elucidation of the mechanism of action of this compound.

## Figures and Tables

**Figure 1 molecules-29-00488-f001:**
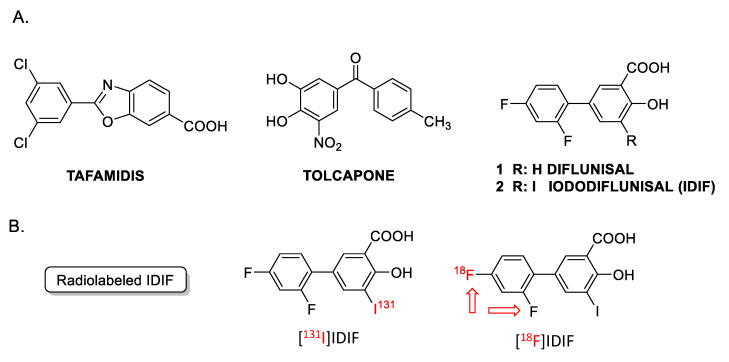
(**A**) Examples of TTR tetramer stabilizers: tafamidis, tolcapone, diflunisal and iododiflunisal (IDIF); (**B**) chemical structure of radiolabeled derivatives of IDIF. The [^18^F]IDIF synthesis has been tackled in this work. (Note: The red arrows indicate that both fluorine atoms can be radiolabeled).

**Figure 2 molecules-29-00488-f002:**
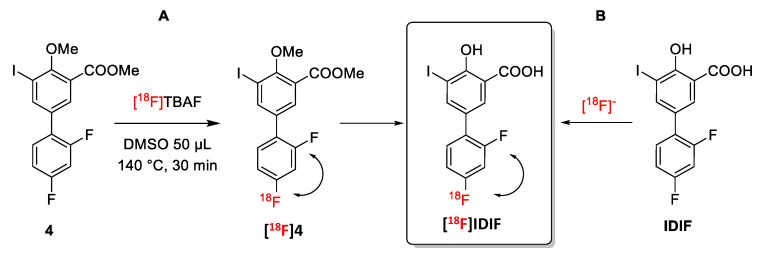
Halogen exchange (^18^F/^19^F) radiofluorination reactions for the preparation of [^18^F]IDIF: (**A**) Using a protected IDIF derivative **4**; and (**B**) directly from IDIF.

**Figure 3 molecules-29-00488-f003:**
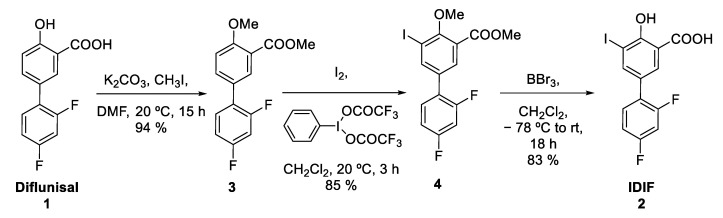
Synthesis of IDIF precursor (**4**) from the NSAID diflunisal (**1**).

**Figure 4 molecules-29-00488-f004:**
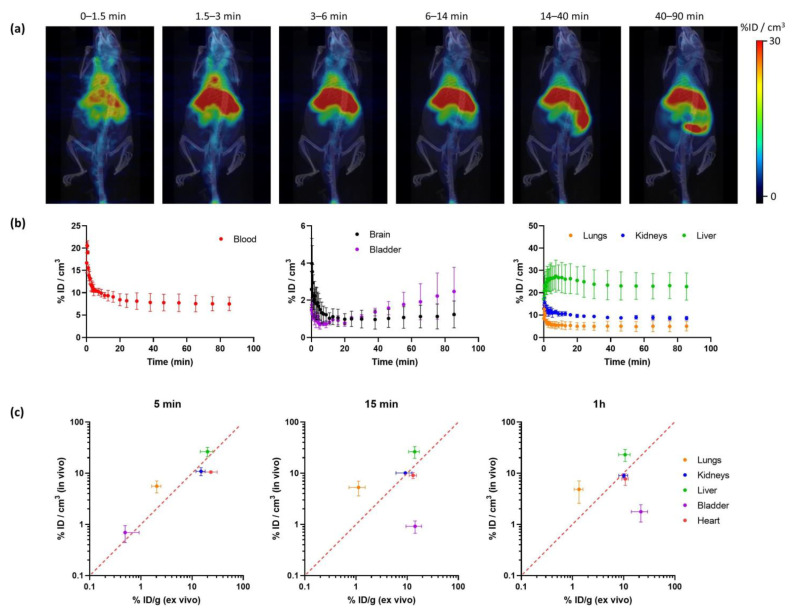
(**a**) Representative PET-CT images obtained at different times after administration of [^18^F]IDIF in wild type mice. PET images correspond to maximum intensity projections, and CT images correspond to 3D-rendered representations. Images have been generated by averaging all frames within each time-window; (**b**) time-activity curves showing the concentration of radioactivity in each organ. Values in the blood are derived from a volume of interest (VOI) drawn in the heart. Values are expressed as percentage of injected dose per cm^3^ of tissue, mean ± standard deviation, N = 3; (**c**) comparison of uptake values in different organs at different time points after tracer administration (5 min, 15 min and 1 h) obtained from in vivo studies using [^18^F]IDIF and PET (current work; values expressed as %ID/cm^3^) and ex vivo studies using [^131^I]IDIF and dissection/gamma counting (previous study; values expressed as % ID/g [33]).

**Table 1 molecules-29-00488-t001:** Optimization conditions for the halogen exchange (^18^F/^19^F) radiofluorination reaction on IDIF as precursor of the [^18^F]IDIF.

Entry ^1^	Solvent	IDIF (mg)	Temperature (°C)	RCC (%)
1 ^2^	DMSO	1	80	0
2	DMSO	4	80	0
3	DMSO	4	120	0.8 ± 0.4
4	DMSO	1	160	12.4 ± 2.9
5	DMSO	4	160	20.6 ± 3.7
6	DMF	4	150	2.2 ± 1.6

^1^ Radiochemical conversion (RCC) values obtained for direct ^18^F/^19^F exchange reaction on IDIF under different experimental conditions, as determined by radio-HPLC. Reaction time: 20 min. ^2^ The number of repetitions per experimental scenario was 3.

## Data Availability

Data are contained within the article and Appendix A.

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
