# Peer review of "Synthesis and PET Imaging Biodistribution Studies of Radiolabeled Iododiflunisal, a Transthyretin Tetramer Stabilizer, Candidate Drug for Alzheimer’s Disease"

_molecules, 2024, doi:10.3390/molecules29020488_

Round 1

Reviewer 1 Report

Comments and Suggestions for Authors

The manuscript by Joshi et al. describes the radiosynthesis of [18F]IDIF via isotopic [19F/18F] exchange, as well as its in vivo evaluation in healthy mice. The manuscript is very well written and has a clear structure throughout. Some aspects, however, need to be addressed before publication.

1.     Line 109: “On one hand, the presence of an iodine atom in the aromatic ring may lead to the 18F/I exchange”. The authors mostly evaluate the radiosynthesis by HPLC, did they synthesise a cold reference of this potential side-product? If not, how do the authors know that this side-product was not formed (it may have a similar elution time as the desired end product).

2.     Line 153: “In all cases, reaction time was fixed at 20 min.” Did the authors test longer reaction times to see if that resulted in improved RCC?

3.     Table 1: Interestingly, the reaction in DMF resulted in a significantly lower RCC compared to the reaction in DMSO. Could the authors briefly elaborate why they think this is?

4.     Regarding the relatively low RCC for isotopic exchange reactions in general, the authors should mention the starting activity (MBq) for the radiosynthesis of [18F]IDIF. Do the authors think that such isotopic exchange reactions could realistically be scaled up for clinical studies (i.e. low yield, high hand dose for the radiochemist, etc.)?

5.     Line 204: “Because the previous studies were performed with [131I]IDIF, the result suggests a good stability of the labeled compounds after administration.” Could the authors please elaborate on this a bit further?

6.     Line 267: “compound 2” should be “compound 4”.

7.     Although the manuscript mostly focuses on the radiosynthesis of [18F]IDIF, biodistribution studies were only performed in healthy mice. Since Alzheimer mouse models are nowadays widely available, the authors should repeat the biodistribution studies in such a disease model. After all, that is the mean reason why the radiosynthesis was initiated in the first place.

Reviewer 2 Report

Comments and Suggestions for Authors

The manuscript submitted by Joshi SM, et al. reports the synthesis and evaluation of F-18 labeled iododiflunisal (IDIF) for imaging with positron emission tomography (PET). IDIF is a transthyretin tetramer stabilizer and is a candidate drug for Alzheimer’s disease. The ability to measure the pharmacokinetics of IDIF noninvasively with PET will facilitate its clinical translation. This reported research is novel, the data is promising, and the conclusions are supported by the presented data. Listed below are some minor suggested changes:

·       The title in the manuscript and the one in the supplementary materials are different.

·       Since biodistribution of I-131 labeled IDIF in mice has been reported (ref 33), the authors should compare the dada of F-18 labeled IDIF with the previously reported data of I-131 labeled IDIF.

·       Table 1: Need to add number of experiment (N = ?) for each entry.

·       Figure 4a: Need to add a color bar and scale.

·       Figure 4b: The uptake in blood and liver is not coming down after 20 min post-injection. Any explanations for the high and retained uptake in blood and liver?

·       Figure 4c: Need to provide a table in supplementary materials for the ex vivo biodistribution data, so others can cite the uptake values if needed.

·       Please follow the published consensus nomenclature rules for radiopharmaceuticals (Coenen HH, et al. Nucl Med Biol 2017; 55: v–xi). For example, [18F]-IDIF should be [18F]IDIF.

·       Page 7: “yielded the desired compound 2 as a white solid (523 mg, isolated yield: 94 %)” should be yielded the desired compound 3 as a white solid (523 mg, isolated yield: 94 %)”.  

·       Page 7: “yielded compound 2 as a yellow oil (369 mg, isolated yield: 85%)” should be yielded compound 4 as a yellow oil (369 mg, isolated yield: 85%)”.

·       The authors should provide MS spectra of compounds 2-4 if available.

·       Page 8: “eluted to the reactor with a solution of Kryptofix K2.2.2/K2CO3 in a mixture of water and acetonitrile”. Need to specify the mass for Kryptofix K2.2.2 and K2CO3, and the volume for water and acetonitrile.

·       Need to provide the procedures for the ex vivo biodistribution study.

Round 2

Reviewer 1 Report

Comments and Suggestions for Authors

I'd like to thank the authors for their detailed responses to my previous suggestions/comments. No further comments.